# Sharp zero-phonon lines of single organic molecules on a hexagonal boron-nitride surface

Robert Smit [1], Arash Tebyani [1], Jil Hameury [1], Sense Jan van der Molen [1] & Michel Orrit [1]✉

Single fluorescent molecules embedded in the bulk of host crystals have proven to be sensitive probes of the dynamics in their nano environment, thanks to their narrow (about 30–50 MHz or 0.1–0.2 µeV) optical linewidth of the 0-0 zero-phonon line (0-0 ZPL) at cryogenic temperatures. However, the optical linewidths of the 0-0 ZPL have been found to increase dramatically as the single molecules are located closer to a surface or interface, while no 0-0 ZPL has been detected for single molecules on any surface. Here we study single terrylene molecules adsorbed on the surface of hexagonal boron-nitride (hBN) substrates. Our low-temperature results show that it is possible to observe the 0-0 ZPL of fluorescent molecules on a surface. We compare our results for molecules deposited on the surfaces of annealed and non-annealed hBN flakes and we see a marked improvement in the spectral stability of the emitters after annealing.

Low-temperature spectroscopy of single-molecules, reaching near to lifetime-limited emission of the purely electronic transition (0-0 zero-phonon line or 0-0 ZPL for short)[1], has so far been exclusively performed on molecules embedded in the bulk of three-dimensional matrices. Such host matrices, for example for terrylene, can be noble gases[2], polymers[3], n-alkanes[4] (Shpol'skii matrices) or aromatics[5–7]. However, molecules that are closer to an interface or surface tend to show anomalous behavior compared to those embedded deeper in the bulk of the material. A study of single terrylene molecules in the hexadecane Shpol'skii matrix reported that molecules closer to an interface with silica were subject to a strong broadening of the 0-0 ZPL, while no 0-0 ZPL was observed for molecules located directly on the silica[8]. Likewise, a study of nanoscopic channels filled with a matrix of dibenzanthanthrene molecules in a naphthalene crystal has shown that a decrease in crystal thickness results in an increased broadening of the 0-0 ZPLs[9]. Lastly, a study on single molecules that had diffused at various depths into a polymer layer, also pointed to a broadening of the 0-0 ZPL with decreasing distance to the polymer surface[10]. Strikingly, the study claimed that no 0-0 ZPL was detected for molecules closer than 0.5 nm to the polymer surface, while molecules that were buried slightly deeper into the polymer (up to a few nm) were subject to intense broadening and rapid spectral diffusion of their 0-0 ZPL due to coupling to two-level systems (TLSs). Beyond single-molecule studies, narrow 0-0 ZPLs down to 500 MHz in width have been observed through hole-burning spectroscopy, e.g. for chemisorbed, and thus strongly bound, quinizarin on alumina surfaces[11], but this particular method relies on weak photostability and thus renders it impossible to detect single molecules. The causes for the anomalous behavior of 0-0 ZPLs of single molecules close to a surface or interface as compared to molecules in the bulk is still under debate. Possibly, high densities of TLSs at or close to the surface have lower activation energies due to the ill-condensed boundaries of a crystal. Impurities could also pile up at the crystal boundaries due to expulsion from the bulk or by adsorption from the environment and contribute to a local increase of conformational states. Despite these surface-induced effects on 0-0 ZPLs, it has been possible to reduce the dimensions of host-guest crystals and yet obtain near-lifetime-limited emission in the bulk of aromatic nanocrystals <100 nm thick, grown by reprecipitation from solution[12,13]. Thinner matrices for single molecules are highly desirable for applications such as single-charge sensing, which requires a close proximity of the sensing molecule to the charge[14], or potential applications in nanophotonics[15]. For the latter case, such systems-on-a-chip

[1]Huygens-Kamerlingh Onnes Laboratory, LION, Postbus 9504, 2300 RA Leiden, The Netherlands. ✉e-mail: Orrit@Physics.LeidenUniv.nl

make use of dielectric waveguides, and a closer proximity of single molecules to waveguides is expected to lead to a better coupling to the evanescent field. The coupling of single molecules to waveguides on a chip was demonstrated in 2017[16]. Later experiments with waveguide-coupled single dibenzoterrylene (DBT) molecules revealed spectral diffusion of the 0-0 ZPL under an applied electric field, attributed to charge fluctuations induced within the nanoguide itself[17]. With high-quality anthracene crystals of about 150 nm in thickness, suspended over the waveguide channels, it was demonstrated that DBT molecules coupling to the waveguide remain stable and near-lifetime-limited for hours[18].

The advent of two-dimensional materials has initiated studies of new classes of single emitters, namely light-emitting defects and color centers[19]. Unlike the aforementioned single molecules, these defects and color centers are covalently incorporated into the host system, and can show relatively sharp zero-phonon lines at room temperature thanks to the rigidity of their host[20]. One such promising 2D material is hexagonal Boron-Nitride (hBN). In comparison to silica, hBN was found to have a much more static environment, leading to stabler emitters hosted by carbon nanotubes that were deposited onto hBN[21]. The lattice structure of hBN itself can host emitters over a surprisingly broad spectral range, extending from the deep ultraviolet[22] up to the near-infrared[23], due to its large band gap of ~6 eV[24,25]. The atomic and electronic structure of many of these emitters still remains a puzzle and only educated guesses as to their origin have been made from quantum-chemistry models[26–29]. Contrary to these emitters inside hBN, single molecules are well-studied and have well-known chemical structures and photophysical properties. A first study at room temperature by S. Han et al.[30], showed that terrylene molecules adsorbed on hBN could be measured for a prolonged time due to a considerable decrease in bleaching rates. In the present work, we searched and found the 0-0 ZPL of terrylene molecules at low temperatures.

We present measurements of the 0-0 ZPLs of single terrylene molecules on the surface of hBN, by high-resolution excitation laser spectroscopy. We reveal remarkably narrow linewidths down to a few 100 MHz, less than a factor 10 away from the lifetime-limited linewidth of about 45 MHz. On a larger scale of GHz's up to THz's, the single terrylene molecules are subject to photo-induced spectral diffusion appearing as spectral jumps, often arising from coupling to two-level systems (TLSs). We observe clear differences between hBN substrates that were or were not annealed, prior to deposition of molecules. In particular, we see a significant reduction in the number of TLSs and in the spectral diffusion after annealing in an oxidizing atmosphere,

which we attribute to the removal of (organic) contaminants from the surface of hBN. Moreover, another spectroscopic site for terrylene is revealed after annealing with a 0-0 ZPL shift of about 20 nm to the red. Remarkably, the spectrum of molecules in this site exhibit strongly reduced vibronic couplings correlated with shifts of their intramolecular vibrations.

## Results

### Localizing single emitters on hBN

Terrylene molecules (see structure in Fig. 2a) were sublimated in vacuum onto multilayer hBN flakes exfoliated onto Si/SiO₂ substrates (see methods). The multilayer flakes were typically <100 nm in thickness (Supplementary Fig. 2). A fluorescence map of the single emitters on hBN is shown in Fig. 1a. Generally, we observe clustering of emitters around sharp lines, which are either step edges or wrinkles. Such clustering is not expected if sublimated molecules land on the hBN surface and immobilize readily. Rather, the landed molecules likely diffuse over the surface before finding a place to immobilize. Interestingly, defect emitters in exfoliated hBN are also preferentially located around sharp edges[31,32]. Isolated emitters show antibunching (Fig. 1b and more in Supplementary Fig. 19) with a dip below 0.5 and an average lifetime of $3.6 \pm 0.2$ ns. This lifetime matches well with the reported lifetime for terrylene on hBN at room temperature: $3.44 \pm 0.38$ ns[30]. For terrylene in general, the lifetime was found to remain constant or to increase slightly between room temperature and liquid-helium temperatures in various matrices[33].

### Identifying terrylene by its spectral fingerprint

At room temperature, the spectra of the emitters are broad (Supplementary Fig. 6) and generally exhibit a main emission peak around 582 nm. This clearly deviates from the observation by Han et al., who reported an emission peak around 600 nm[30]. An emission around 600 nm would correspond to a significant red-shift when compared to terrylene in organic matrices, (with the exception of *p*-dichlorobenzene, where the red-most spectroscopic site of terrylene was found at 597 nm[6]). However, by sublimating terrylene on annealed hBN flakes, we also find a small sub-population of molecules with an emission peak around 602 nm, with a spectrum that differs significantly from that of the molecules in the 582 nm site (compare Fig. 2c, d).

At liquid-helium temperatures, the spectra have narrowed down to the limit of the spectrometer's resolution ($1.7 \pm 0.1$ cm⁻¹). This allowed us to identify each emitter independently by its vibrational

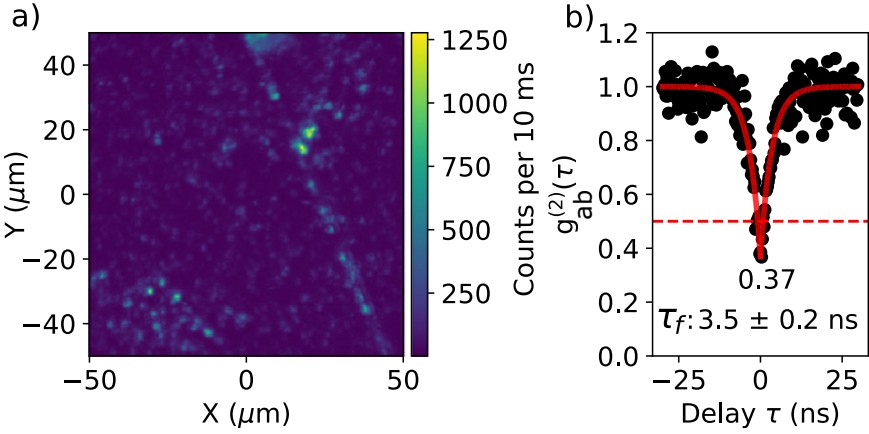

**Fig. 1 | Identifying single emitters on hBN. (a)** Confocal-fluorescence intensity map of a large (non-annealed) hBN flake showing single spots and clusters of terrylene molecules that were excited by a vibronic transition with 532 nm light. **b** Antibunching histogram recorded from a typical isolated spot and normalized to unity, neglecting contributions from photon bunching at longer time scales. The dip does not fully extend to zero due to a relatively strong background in the fluorescence signal. This background was caused by leakage through the filter (532 nm Notch) of the high-intensity excitation light (17 kW/cm²), together with a weak Raman signal from Si (520 cm⁻¹) and hBN (1365 cm⁻¹). The red curve is a fit of the function $g_{ab}^{(2)}(\tau) = 1 - ce^{-|\tau|/\tau_f}$ to the data points.

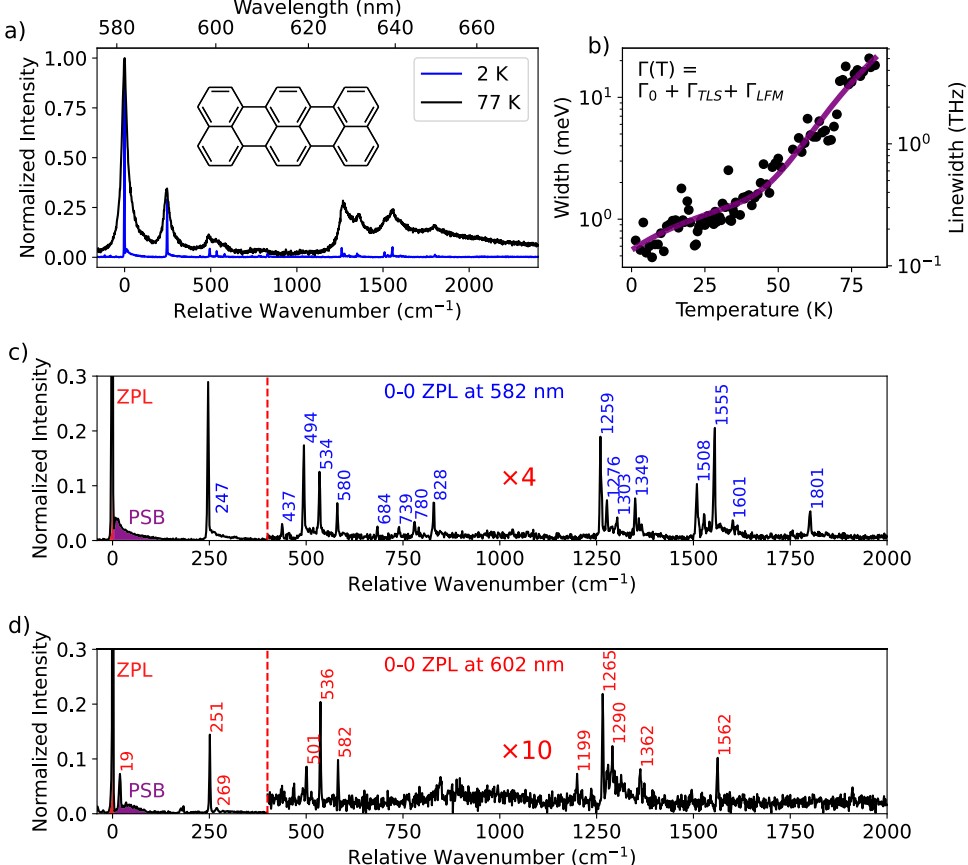

**Fig. 2 | Fluorescence spectra of single terrylene molecules and the effects of temperature. a** Emission spectra of a single terrylene molecule around 582 nm, taken at two temperatures, 2 K (blue) and 77 K (black). All spectra have been normalized to unity relative to the intensity of the 0-0 zero-phonon line (ZPL). **b** Linewidth of the 0-0 ZPL of a single terrylene molecule (not the one shown in (**a**)) on annealed hBN (750 °C for 12 h). The linewidth was fitted to the equation in inset, where $\Gamma_{TLS}$ scales linearly with temperature and $\Gamma_{LFM}$ is given by the Arrhenius law, described in the main text. The linewidth at the lowest temperature was limited by the spectrometer resolution ($1.7 \pm 0.1\,\mathrm{cm}^{-1}$). Some spectra of this molecule can be found in Supplementary Notes 3. **c** Zoomed-in version of the spectrum at 2 K in (**a**)

with vibrational peaks annotated by their energy in $\mathrm{cm}^{-1}$. The part of the spectrum after $400\,\mathrm{cm}^{-1}$ has been magnified by a factor 4. **d** Details of a spectrum of a molecule with 0-0 ZPL located at 602 nm. Note that the vibronic lines are quite weak compared to those in the spectrum of (**c**) requiring for visibility a magnification factor of 10 for the part after $400\,\mathrm{cm}^{-1}$. The Debye-Waller factors, which are known as the intensity of the 0-0 ZPL with respect to the combined intensity of the 0-0 zero-phonon line (ZPL, red area) and the phonon sideband (PSB, purple area), thereby excluding intramolecular vibrational peaks, are around $0.7 \pm 0.1$ for the molecule at 582 nm and $0.8 \pm 0.1$ for the molecule at 602 nm. The peak around $19\,\mathrm{cm}^{-1}$ in (**d**) was included in the phonon sideband.

fingerprint (Fig. 2a with a detailed version in Fig. 2c). The spectrum in Fig. 2c clearly displays the fingerprint of terrylene, which has been studied before in great detail in various matrices[34–36]. These assignments of the vibrations and comparisons of terrylene in other matrices can be found in Supplementary Notes 2. The coarse vibrational fingerprint could also be resolved at 77 K with relative ease (Fig. 2a), which makes the use of liquid helium not absolutely necessary for the identification of the emitters. In cases where the samples were prepared by spin coating of molecules in a toluene solution, instead of the sublimation method mentioned above, impurity emitters could be found regularly (Supplementary Notes 7). These emitters have 0-0 ZPLs spread over a broad range of 618 nm up to 640 nm and exhibit the fingerprint of a well-known but unidentified impurity of polymers[37] and solvents[38], such as toluene[39]. To our knowledge, no reports of this impurity on hBN have been published before, but we would like to warn readers about the possibility of wrongly assigning this impurity to intrinsic emitters of hBN.

## Broadening of the zero-phonon lines with temperature

We characterized the effect of temperature on the emission spectrum of a single terrylene molecule by the broadening of the 0-0 ZPL. We measured the linewidth or full-width-at-half-maximum for 88

intermediate temperatures in the range from 1.4 K up to 83 K. The 0-0 ZPL linewidth follows a broadening relation ($\Gamma(T) = \Gamma_0 + \Gamma_{TLS} + \Gamma_{LFM}$) that is consistent with defects in hBN[40,41] or disordered matrices, such as polymers[42]. This similarity arises from the thermal activation of tunneling two-level systems (TLSs), for which the contribution to the broadening follows a quasi-linear relationship with temperature: $\Gamma_{TLS} = bT^{\alpha}$, with $\alpha \approx 1$. The exponential term is accounted for by the population of quasi-localized low-frequency modes (LFMs), which follows the Arrhenius law: $\Gamma_{LFM} = w \times \exp(-E_a/k_bT)$. Here $E_a$ is defined as the activation energy of the LFMs. Although there is not much literature on the linewidth of single molecules in 3D matrices up to such high temperatures, recorded by emission spectra, one work mentions a linewidth of a single terrylene molecule in Shpol'skii matrices $n$-hexadecane and $n$-dodecane of respectively $33 \pm 3\,\mathrm{cm}^{-1}$ and $21 \pm 3\,\mathrm{cm}^{-1}$ at 50 K[43]. This compares well to a linewidth of about $25\,\mathrm{cm}^{-1}$ at 50 K for the terrylene molecule on hBN, making this system very comparable to the bulk of a Shpol'skii matrix at higher temperatures. However, the contribution from TLSs, estimated as $21 \pm 14\,\mu\mathrm{eV/K}$ ($5 \pm 3.5\,\mathrm{GHz/K}$) seems to be considerably larger than the typically less than $100\,\mathrm{MHz/K}$ for Shpol'skii matrices[42], meaning that these systems probably broaden quicker at lower temperatures. The stronger TLS broadening might be due to the closer proximity of these TLSs, as they are

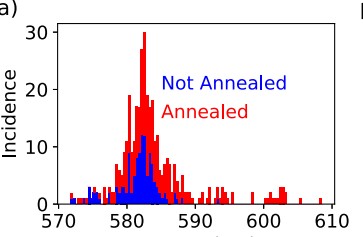 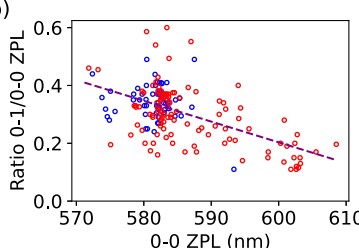 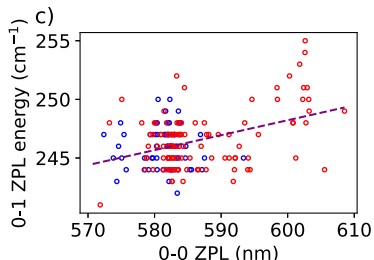

**Fig. 3 | Statistical study of single terrylene molecules on annealed and non-annealed hBN. a** 0-0 zero-phonon line (ZPL) positions of 373 terrylene molecules divided into sets of two colors. The blue points (123 molecules) represent 0-0 ZPL positions on non-annealed hBN flakes, while the red points (250 molecules) correspond to flakes subject to annealing with at least 500 °C and for some samples up to 900 °C. After annealing, a small and narrower distribution of 17 molecules is found around 602 nm with some additional molecules between 582 nm and 602 nm. **b** Shows the measured relative intensity of the main vibration (0-1 ZPL) versus the position of the 0-0 ZPL for 58 molecules on non-annealed hBN (blue points) and 113 molecules on annealed hBN (red points). On average, the more red-shifted molecules have a weaker vibronic coupling (Pearson's correlation coefficient $r = -0.56$). For the same set of molecules, a less-significant relation ($r = 0.4$) is found for the vibration energy compared to the 0-0 ZPL position in (**c**). The statistics in (**a**) are higher than for (**b**, **c**) because the ratio and energy of the vibrational peaks were not always clearly resolved in the spectra due to overlap with other molecules.

probably related to the (organic) contamination on the surface. Beyond the linear regime, there is a clear exponential take-off from about 50 K, which corresponds to a rather high Debye temperature of $405 \pm 41$ K. This is considerably higher than the 10–40 K mentioned for dibenzoterrylene (DBT) in anthracene[44-46] and found in all organic matrices. However, DBT molecules in anthracene broaden with a pure Arrhenius curve and do not suffer from additional broadening of the homogeneous linewidth due to TLSs.

### Distinct spectroscopic sites for terrylene on hBN

We find the 0-0 ZPLs of terrylene molecules to be scattered over a broad range, though with clustering into specific spectroscopic sites (Fig. 3a). The distribution around 582 nm—which we will call the main site—displays a relatively broad inhomogeneous broadening of about 4 nm (120 cm⁻¹), which is typical for disordered systems such as polyethylene[3]. On hBN flakes annealed prior to molecule deposition, we also find a new site with a relatively narrow distribution around 602 nm, which we will call the red site. Surprisingly, the molecules in this site show a significantly reduced vibronic coupling (Franck-Condon factors) between the excited state and vibronic levels of the ground state. Overall, we observe a decrease of the Franck-Condon factors when the 0-0 ZPL shifted to the red (Fig. 3b). In addition, there appears to be a weak positive relation between the 0-0 ZPL position and the energy of the main vibration (frequency difference between the 0-1 ZPL and the 0-0 ZPL), shown in Fig. 3c. We did not detect any single molecule in the red site without prior annealing of the hBN flakes. Annealing, at the temperatures we applied (500 °C up to 900 °C) is expected to remove most organic contamination[47] and to redistribute defects such as vacancies[48], as the removal of structural defects would require much higher temperatures, up to 1700 °C[49]. Remarkably, we find considerably fewer molecules on annealed samples, although we kept the sublimation rate of terrylene fixed or even slightly increased it (Supplementary Fig. 3). We propose as a possible explanation that terrylene anchors to (organic) contamination at the hBN surface. As annealing dramatically reduces the concentration of anchors, terrylene molecules will either leave the flake area or may aggregate to the few nucleation sites and stop fluorescing because of self-quenching. Another possibility is that the annealing process creates dangling bonds at defect sites, which may react with terrylene upon sublimation and bleach them in result. Interestingly, we find that the number of terrylene molecules on the flake increases again if prior to molecule sublimation, the annealed samples are intentionally contaminated by the spin-coating of *n*-hexadecane, which is known to form monolayers on top of hBN[50]. On the one hand, contamination could help terrylene to find more anchors and favor immobilization. On the other hand, contamination could also prevent terrylene from

finding a site where it can interact strongly with the hBN, for example with some defect in hBN. We speculate that these hBN defects might be responsible for the red-shifted molecules. They would be accessible only after annealing of the hBN, right before the terrylene molecules are sublimated. The need for anchoring points could possibly be avoided by in-situ evaporation of terrylene on cold hBN surfaces, which is not possible in our setup at this moment.

### Spectral diffusion

We find that annealing of the hBN flakes improves the spectral stability of the single emitters. We assign the spectral diffusion to jumps of (tunneling) two-level systems (TLSs). This observation is consistent with our hypothesis that these TLS's may have been located in the organic contaminants. The improvement of the spectral stability is most clear for an annealing temperature between 500 °C up to 750 °C, but much less clear for higher temperatures, up to 900 °C. An evident case of coupling of a molecule to a single TLS is shown in Fig. 4a, which traces the 0-0 ZPL position over time at gradually-increased excitation intensities. In Fig. 4b, the number of spectral jumps observed in a time window of 200 s is related to the power density of the laser spot at the position of the molecule and follows the same relation as the fluorescence intensity of the 0-0 ZPL. Therefore, these spectral jumps are laser-induced in one-photon processes. Later, we will show that spectral jumps are also observed by resonant excitation of single molecules, which requires power densities of at least three orders of magnitude weaker, due to the narrow width of the 0-0 ZPL with respect to the width of a vibronic transition.

We consider several possible sources as a cause for the spectral diffusion. One of them is a spatial jump of the molecule itself. However, rotation of the molecule can be ruled out by analysis of the fluorescence polarization (Supplementary Fig. 13), while super-resolution imaging did not reveal an obvious translational movement either (Supplementary Figs. 14, 15). Clearly, the number of two-level systems is reduced significantly after annealing, while also the amplitude of spectral diffusion is observed to decrease (Fig. 4a, d, e). Without annealing, the spectral jumping is in general much more complex, consisting of many, possibly coupled, levels, whose population rates could change over time (Fig. 4c). The scale of the spectral jumps, extending up to a few THz in some cases, and the lack of correlation between spectral diffusion of emitters in the same focal area, points to events in the close vicinity (few nm) of the molecule. As discussed before, the molecule is likely anchored to (aggregates of) contaminants on the surface. This (nonfluorescent) contamination itself can be responsible for the spectral jumps, i.e. by (a group of) atoms tunneling between two spatial positions, perturbing the optical transition of the terrylene molecule by electrostatic or elastic dipole-dipole

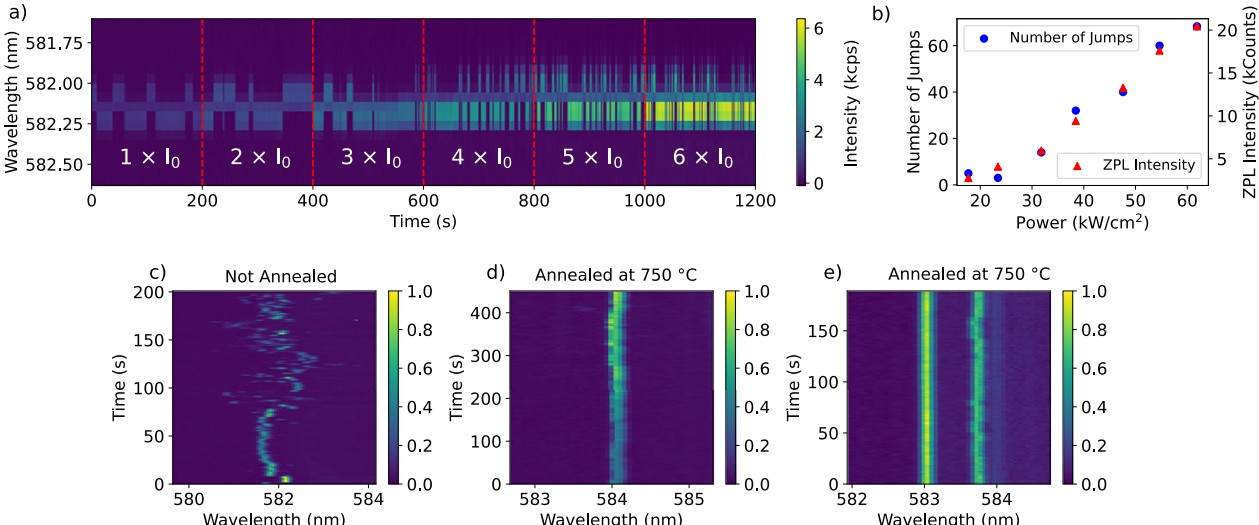

**Fig. 4 | Coupling to two-level systems and the effects of annealing on emitter stability. a** A 0-0 zero-phonon line (ZPL) of a terrylene molecule in the main site followed for 20 min on an hBN flake that was annealed at 750 °C for 12 h, excited by a vibronic transition with 532 nm light. The excitation intensity is raised after each frame of 200 s. The associated power densities, relative to the initial intensity $I_0$, are shown in (**b**), together with the number of spectral jumps that are observed in the 200 s time window and the integrated intensity of the 0-0 ZPL for a period of 1 s. Without annealing, the spectral time trace typically exhibits complex spectral diffusion as shown for example in (**c**). Initially, the molecule wanders around a relatively small spectral region, but extends over a much wider region from about 70 s. After annealing, the amplitude of spectral diffusion is typically strongly reduced, as in (**d**), even though the laser power is doubled at each 100 s interval. Some molecules do not show any spectral diffusion on the scale resolved by the spectrometer, as is the case for the leftmost molecule in (**e**). The spectra in (**c**) are measured on a different sample than the spectra in (**d**, **e**). More examples of spectral traces can be found in Supplementary Fig. 18.

interaction. However, the exact nature of these TLSs remains unknown due to the random environment around the molecule.

### Resonant excitation of the 0-0 ZPL

In the data presented so far, single molecules were excited through a vibronic transition and not resonantly through the 0-0 ZPL. The 0-0 ZPL linewidth resolved in the emission spectrum is limited by the spectrometer resolution ($51 \pm 3$ GHz per pixel), while the linewidth of terrylene is expected to be up to 3 orders of magnitude narrower, around $45 \pm 3$ MHz, if it is limited by the fluorescence lifetime of $3.6 \pm 0.2$ ns. With a tunable dye laser (linewidth of a few MHz) we excited single terrylene molecules resonantly. In many cases, the molecule jumped out of resonance with the excitation laser, already in the first scan. With lower excitation intensities of a few W/cm², we could follow the molecules longer. Again, the most stable molecules were found on the annealed samples.

A distribution of linewidths of the 0-0 ZPL of single terrylene molecules is shown in Fig. 5c. The narrowest 0-0 ZPLs are about a factor 10 above the lifetime-limited linewidth. The median of the distribution lies around 1 GHz (4.1 μeV). This is not very different from defect emitters reported for hBN, which show similar linewidths of the ZPL under resonant excitation[41]. In rare cases, we find molecules that did not move out of the scanned frequency range or only for a very short period. One example is shown in Fig. 5a (others in Supplementary Notes 6). This molecule is measured on an annealed flake and has a relatively broad linewidth of $4.7 \pm 0.3$ GHz and a saturated count rate of about $79 \pm 6$ kcps, obtained from the saturation curve in Fig. 5d. The molecule is stable, although the asymmetric shape of the 0-0 ZPL might indicate some spectral diffusion to one side.

We show a fluorescence time trace with resonant excitation of that molecule in Fig. 5b and it displays so-called quantum jumps in the fluorescence signal due to intersystem crossing to the triplet state[51]. To analyze the characteristic time scales of these quantum jumps, we determined a threshold between the ON and OFF state, where a crossing of this threshold would indicate a change from ON to OFF or vice versa. For a time trace of 60 s, the lengths of the periods where the

molecule was OFF are plotted into a histogram, shown in Fig. 5e. The histogram is best fitted by a bi-exponential decay, with characteristic triplet lifetimes of $360 \pm 10$ μs and $2.0 \pm 0.1$ ms. We attribute the short decay time to the indistinguishable decays of the in-plane triplet states $T_{xy}$, while the remaining long decay corresponds to the out-of-plane state $T_z$[52]. The triplet lifetimes agree well with terrylene in the extensively studied hosts *p*-terphenyl[7,51,53] and anthracene[54]. Despite the general agreement with the bi-exponential fit, the existence of datapoints at considerably longer times may point to another source of fluorescence blinking, possibly caused by relatively short-lived spectral jumps. This other source of blinking is also resolved in the autocorrelation function in Supplementary Fig. 24b. We continued by recording an antibunching histogram of the resonance fluorescence (Supplementary Fig. 24a). No Rabi oscillations are observed in the histogram, which is expected when the linewidth is significantly broadened by dephasing. The measured linewidth of $4.7 \pm 0.3$ GHz would correspond to a lower bound of the decoherence time $T_2^*$ of about $68 \pm 4$ ps.

### Discussion

We have shown that terrylene molecules adsorbed on the surface of hBN become narrow emitters at low temperature, with linewidths as narrow as a few 100 MHz up to a few GHz, which is similar to intrinsic hBN defects[40,41,55,56]. Their relative spectral stability, in contrast to single molecules adsorbed on any other surface so far, made it possible to observe 0-0 ZPLs on a surface. Moreover, we have found a way to considerably improve the spectral stability by annealing the hBN substrates before the molecules were deposited, which points to (organic) contamination as a potential source of spectral jumps. Spectral instabilities for emitters hosted by hBN are not new and are also observed for defect emitters inside hBN, which become particularly prominent at low temperature[27,40,41,56–58]. In future experiments, a well-known system such as terrylene could help shed light on these various issues of spectral instabilities and dephasing mechanisms, in order to improve the quality of emitters hosted by hBN. Moreover, the sensitivity of the molecules to their surroundings shows that our

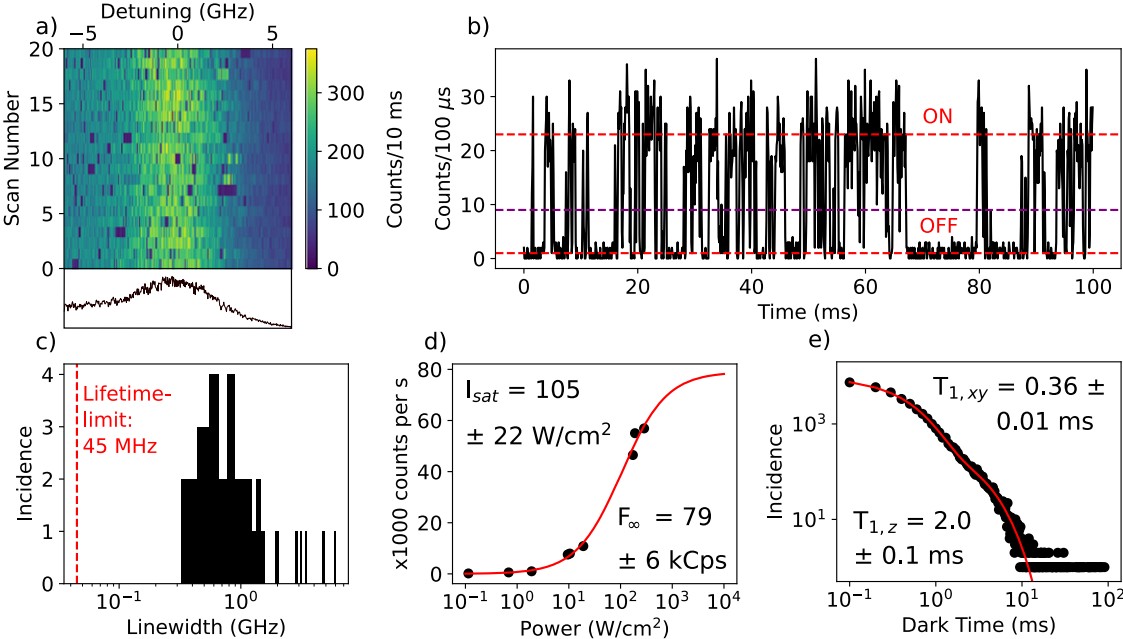

**Fig. 5 | Properties of resonantly excited molecules. a** Excitation spectrum of the 0-0 ZPL of a single molecule that did not jump out of resonance with the laser, even at higher excitation intensities. The linewidth is about $4.7 \pm 0.3$ GHz and is measured with a scan rate of 5 s per row. The line plot underneath the figure shows the asymmetric profile of the 0-0 ZPL. A resonance fluorescence time trace of the fluorescence signal of this molecule in (**b**) shows characteristic quantum jumps due to intersystem crossing to the triplet state. However, in some cases the molecule remained dark for much longer times than the triplet lifetime, as is also visible in (a). These long dark times may be attributed to a relatively short spectral jump back and forth from and to another (out-of-range) spectral position. The purple line represents the threshold set between the ON and OFF state of the fluorescence. **c** Distribution of 0-0 ZPLs linewidths found for molecules on different flakes. The

red dashed vertical line represents the lower limit set by the fluorescence lifetime. Due to the large distribution of linewidths, the horizontal axis has been set to a logarithmic scale. **d** Saturation curve of the fluorescence of the molecule in (**a**), fitted with $F(I) = F_\infty \frac{I/I_{sat}}{1+I/I_{sat}}$, where $I_{sat}$ is the saturation intensity of the laser and $F_\infty$ is the maximum fluorescence rate. The saturation intensity is $105 \pm 22$ W/cm², 2 orders of magnitude larger than typically obtained with our setup for near-lifetime-limited emitters. This is explained by the 2 orders of magnitude broader linewidth, as the saturation intensity scales linearly with the linewidth of the transition. **e** Distribution of dark periods in the resonance fluorescence signal, recorded over 60 s with 100 μs time bins (as in (**b**)). The characteristic timescales of the fit (red curve) are shown on the top right and bottom left.

results are relevant for the study of molecule-surface interactions as well as the study of physical phenomena taking place at the surface of hBN.

To our surprise, we also found a new spectral site after annealing, with a significantly altered spectrum compared to molecules in the main site. The shifts of the vibronic lines and their weak vibronic coupling could be the result of strong interaction with hBN, possibly with some defects. We hypothesize that the reduced amount of organic contamination frees these adsorption sites for single terrylene molecules.

Despite the rigidity of the hBN host, the molecules do not benefit yet from a narrower 0-0 ZPL at higher temperatures. Contrary to emitting defects, which can be buried deep inside the multilayer hBN, the single molecules are likely much more sensitive to dynamics at the surface, leading to a stronger broadening with temperature by the activation of two-level systems. The compatibility of single molecules and hBN paves the way for further studies that could implement the encapsulation of single molecules between hBN layers. We anticipate that the passivation of the molecule's surrounding by one or more hBN layer(s) could further improve their spectral stability and could help bring their 0-0 ZPL linewidth closer to the lifetime limit, even at temperatures higher than those of liquid helium.

## Methods
### Sample preparation
Flakes of hBN from a single crystal (HQ Graphene) were transferred to the substrate by the exfoliation method, where the layers are cleaved using scotch tape. The substrate for hBN was a Si wafer (University Wafer) coated with a 300 nm oxide layer. In early experiments, the

exfoliated samples were cleaned in acetone to remove any residue left from the tape. In later experiments, the cleaning step after exfoliation was omitted, as it might rather contaminate than clean the freshly cleaved hBN flakes. Optical inspection showed that the multilayer flakes were present on the substrate and their lateral sizes varied from a few μm up to a few 100 μm. Annealing of the hBN was performed in a tube oven (Thermcraft) in a moderate vacuum of about $10^{-2}$ mbar of residual air.

Terrylene (synthesized by Mercachem) crystals were placed on the bottom of a sublimation apparatus (Supplementary Fig. 1). The substrate with the hBN flakes was suspended by carbon tape and was in thermal contact with the cold finger that was cooled by water ice. The atmosphere was pumped to vacuum and the flask was heated on a hot plate for a sublimation time of 5 min. A heating temperature of the terrylene crystals between 120 °C and 140 °C was found to yield a suitable concentration of terrylene molecules on hBN (Fig. 1a). Alternatively to the sublimation method, we also employed spin coating as a method for the deposition of molecules. In that case, terrylene was dissolved in toluene (Acros Organics, 99.85%) and diluted to a concentration below 1 nmol/mol. A few droplets of the terrylene solution (around 10−50 μL, depending on substrate size) were pipetted onto the substrate. Spin-coating followed at 2000 rpm for 20 s and was terminated by a drying step at 4000 rpm for an additional 20 s.

### Measurement setup
The samples were fixed to a sample holder and inserted into a flow cryostat (Janis SVT-200-5) that can cool down to 1.2 K. Before cooling down, the cryostat was purged three times by pumping out all the gases inside and exchanging them for dry nitrogen gas. The purging

did not seem to have any noticeable effect on the exposed terrylene molecules. The cryostat contains an objective (0.85 NA, Edmund Optics) that is immersed in liquid helium and forms part of a home-built confocal setup. For the spectroscopy experiments, two excitation sources were used. For a relatively broadband excitation, a 532 nm laser (Sprout G-15W, Lighthouse Photonics) was used, which is phase-locked with six longitudinal modes spanning over 4.25 GHz. As a narrow and tunable excitation source (~1 MHz linewidth) a Coherent 699 dye ring laser was used, which was operated with Rhodamine 6 G dye and pumped by a Coherent Verdi V2 laser (5 W at 532 nm). The wavelength of the dye laser was monitored with MHz precision using a High Finesse WS6-200 wavemeter. The emission spectra were recorded using a Horiba iHR320 spectrometer, which was coupled to a liquid-nitrogen-cooled Symphony II CCD detector. The spectrometer can be operated with three diffraction gratings with 150, 600 and 1200 lines/mm, yielding a spectral resolution down to $1.7 \pm 0.1$ cm-1 around a wavelength of $580 \pm 10$ nm. Confocal fluorescence imaging was performed using a scanning mirror (Newport, FSM-300-01) and fluorescence was detected with one or two APDs (Excelitas, SPCM-AQRH-16). The signal from the two APDs was time-correlated using a PicoHarp 300 from PicoQuant in combination with a PHR-800 router. A programmable delay was set on the stop channel by a delay box (Ortec DB463).

## Data availability

The data generated in this study is available on Figshare[59] under accession code https://doi.org/10.6084/m9.figshare.24213018. Source data are provided with this paper.

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

## Acknowledgements

We thank research engineer Harmen van der Meer for his help with the repair of our cryostat. Furthermore, we acknowledge the AFM Facility of the Leiden Institute of Physics (LION) for enabling the AFM measurements. We also thank in particular Dr. Zoran Ristanovic and Dr. Amin Moradi and all other past group members that have contributed to the build-up of our setups, which have made this work possible. Lastly, we are grateful for the funding of this research, which was provided by the NWO (Spinoza prize 2017).

## Author contributions

R.S. and M.O. conceived the idea for this experiment. A.T. and R.S. performed the sample preparation and the measurements. J.H. was involved in the early experiments of identifying single emitters and recording their spectra. M.O and S.J.v.d.M. contributed to the theoretical understanding of the work.

## Competing interests

The authors declare no competing interests.
