## [Peer Review File · Nature Communications]

Sharp zero-phonon lines of single organic molecules on a hexagonal boron-nitride surfaceREVIEWER COMMENTS

Reviewer #1 (Remarks to the Author):

The manuscript by Smit et al. reports spectroscopic studies of single fluorescent terrylene molecules on hBN substrates subjected to different treatment prior to molecule adsorption. At low temperatures, they find two classes of emitters with spectrally narrow zero-phonon lines and study their blinking and spectral wandering behaviors as a function of time and excitation power. The authors attribute the strong reduction in spectral wandering to an electrostatically calm environment provided by hBN, and reciprocally highlight the potential of single-molecule spectroscopy for optical sensing.

The work is of extremely high quality, both experimentally and conceptually, and the analysis is elaborate, convincing and sound. In my opinion, the manuscript deserves publication in Nature Communication due to its pioneering character, scientific quality and in part also as a reference for ubiquitous single-emitter studies in hBN which might be compromised by organic fluorophores (see also <https://arxiv.org/abs/2306.12197>).

I have only a few minor questions and comments:

1) The authors point out the perspective of both-sided encapsulation of molecules by hBN. I also thought this could be a very instructive experiment to carry out. How much of an extra effort would this work add?

2) Related work on single luminescent nanotubes on hBN (Noe et al, Nano Lett. 18, 4136, 2018) found increased spectral wandering in the proximity of hBN terrace steps, just as they appear in Fig. 1a due to molecule agglomeration. Is there any difference in the spectral characteristics of molecules near the terrace steps and further away? And could the spectral wandering (induced by fluctuating proximal two-level systems) be related to high density of dangling bonds of the hBN terrace edges as opposed to rather low density of vacancy defects on extended terraces?

3) The blinking behavior as in Fig. 5b should yield strong bunching in auto-correlation measurements as in Fig. 1a and S6.1 on time scales of the characteristic switching time between the "on" and "off" states (as e.g. in Nutz et al, Nano Lett. 19, 7078, 2019). Can the authors expand their time window in the auto-correlation data? This on the one hand might yield a different normalization constant for $g^2(\tau)$, and on the other hand might help to clarify the role of triplet states vs. alternative sources of blinking and thus instructively complement the data in Fig. 5e.

Reviewer #2 (Remarks to the Author):

The paper from Smit et al. submitted to Nature Communications details experiments carried out to investigate the optical properties of terrylene molecules deposited on hexagonal boron nitride (hBN). The authors find that molecules can have narrow linewidth transitions even though they are not included inside a protective crystalline matrix. This is significant and novel as molecules on surfaces usually have very broad emission. They find that the molecule emission is more stable when the hBN substrate has been annealed at high temperature. This could be due to the removal of organic contaminants, or some kind of surface passivation. The authors explore the emission spectra of molecules, as well as anti-bunching measurements and laser fluorescence excitation spectra. They compare the molecule emission at various temperatures and observe and analyze broadening effects. Whilst I find the work very interesting, I am unsure whether it is of broad interest nor is it clear whether there are specific applications the authors have in mind which this combination of materials would be useful for. The authors mention "for investigation of physical phenomena on surfaces" which I think they could expand on significantly. They also mention for "nanophotonic devices" but it is not clear how one would combine this hBN with molecules and a waveguide or cavity. The authors also point out that encapsulation in hBN would also be a route to achieving narrower lines, but it is not clear whether this would be inherently better than other more well explored crystalline matrices used to host organic molecules.

I have the following more specific comments:

1. The authors mention that waveguide coupling of molecules was first seen in Shkarin et al (Ref. 15) but previous demonstrations from the same group were made by Türschmann et al., Nano Lett. (2017) and later with ring resonators by Rattenbacher et al., New J. Phys. (2019), as well as demonstrations of coupling to a gap between waveguides by Boissier et al., Nat. Comms. (2021).
2. UV and NIR are not defined.
3. Was the data in Fig. 2b with annealed or not annealed hBN?
4. DBT in anthracene temperature broadening has been analyzed in other papers which give activation energies similar to those considered in the text. See Grandi et al., PRA (2016) and Clear et al. PRL (2020).
5. It would be good to make the red points in Fig. 3 slightly transparent so the distribution of the blue points behind can be seen.
6. How fast were the scans taken in Fig. 5a?
7. Figure 5a would benefit from having a plot that sums all the scans and plots than sum vs detuning to elucidate the asymmetry mentioned in the text.
8. Why does Fig. 5d stop before the curve saturates? It would be better to at least plot this further to show the top of the S-curve even if no more data was taken.

Reviewer #3 (Remarks to the Author):

The paper by Smit et al. reports about the first observation of narrow zero-phonon-lines (ZPL) of single dye molecules adsorbed on the surface of hexagonal boron-nitride (hBN). To conduct the experiments, the authors use the aromatic hydrocarbon terylene, which over more than two decades has allowed a number of "firsts" in single molecule spectroscopy. The detection of single molecules on a surface under ambient conditions - given sufficient photostability - is relatively straightforward and recently has been shown for the same system, i.e., terylene on hBN. While in this latter study the very surprising observation has been the unusually high photostability of single terylene molecules on a surface in the presence of oxygen, in the present study the novel and intriguing result is the occurrence of narrow single molecule ZPLs on a surface. So far, recording of low temperature ZPLs of surface-adsorbed single molecules has remained elusive, mainly because of additional broadening processes and/or increased spectral diffusion. It appears that the peculiarities of the hBN substrate has given rise to a number of unexpected results, truly remarkable.

Smit et al. characterize the single molecules by well-accepted procedures as fluorescence spectra and triplet lifetimes which indicate the presence of terylene. Single molecule behavior is verified by photon-antibunching as seen in the fluorescence correlation function which also yields the fluorescence lifetime expected for terylene. Given the numerous emissions reported from hBN, these are important points. The ZPL linewidth - extracted from excitation spectra - is still appreciably broadened with respect to the lifetime limit which seems to be one characteristic of surface-adsorbed molecules. Overall, the authors present novel and interesting findings which may have a strong impact on further single molecule research and for quantum photonic applications. It is to be expected that also other dye molecules will exhibit single molecule ZPLs on hBN, maybe with even more favorable properties. Moreover, such studies may give an answer to the pending question: What are the intrinsic properties of hBN and/or its surface, respectively, which allow the observation of narrow ZPLs?

The paper is well-written, and all relevant information and procedures are given. I suggest publication in Nature Communications after the authors have considered the following questions and comments.

In the introduction and at several other instances the authors mention that their results would be the first in which 0-0-ZPLs have been observed for fluorescent molecules on a surface. This statement is certainly correct for the important case of single molecules, but not in general. Indeed, 0-0-ZPLs have been observed by several groups by low temperature spectral hole-burning of surface-adsorbed dye

molecules. To give an example (Chem. Phys. Lett. 118 (1991) 179), spectral holes with a width of about 1 GHz have been reported which translates into a ZPL width of 0.5 GHz. While the surface which has been studied was quite different from that of hBN, the ZPL width in this bulk experiment was even narrower than the median given in the present paper. Since (efficient) spectral hole-burning is a process which typically will prevent the detection of stable single molecule ZPLs, the materials used in former experiments will not work for ZPL single molecule detection. Nevertheless, the referee believes that the paper would profit considerably by a critical discussion of the former results, in particular with respect to the general question of linewidth broadening processes on surfaces.

The authors have measured the temperature dependence of the 0-0-ZPL width of a single molecule as obtained from emission spectra. As mentioned by the authors, these suffer from experimental resolution limits at low temperature. Have the authors also tried to measure the temperature dependent linewidth as obtained from excitation spectra which would not suffer from resolution limits? Probably, spectral instabilities make this a difficult endeavor, but for some molecules (Figure 5 (d)) even a saturation curve could be taken. Please comment. By the way, I think the saturation curve has not been discussed in the main text.

The authors mention that after annealing considerably less molecules have been found and present several tentative explanations. To the referee's opinion, they also should take into account that the heat treatment (500 K to 900 K) of their samples can easily lead to reactive sites like the well-known dangling bonds (unsaturated valencies). At these reactive sites, terrylene can be readily decomposed. Such an explanation would also be in line with the observation of more molecules again after treatment with hexadecane which will saturate the reactive sites.

In the introduction the authors use micro-eV and MHz for linewidth information. I think it would be better for the reader not to change the units in the introduction.

We thank the reviewers for their positive reception of our manuscript and their helpful comments that will improve the quality of our report. Hereby, we provide a point-by-point response to their comments and we address the changes that we have applied to the manuscript.

Actions undertaken in response to the reviewer comments are indicated in red.

Reviewer #1

Remarks to the author:

The manuscript by Smit et al. reports spectroscopic studies of single fluorescent terrylene molecules on hBN substrates subjected to different treatment prior to molecule adsorption. At low temperatures, they find two classes of emitters with spectrally narrow zero-phonon lines and study their blinking and spectral wandering behaviors as a function of time and excitation power. The authors attribute the strong reduction in spectral wandering to an electrostatically calm environment provided by hBN, and reciprocally highlight the potential of single-molecule spectroscopy for optical sensing.

The work is of extremely high quality, both experimentally and conceptually, and the analysis is elaborate, convincing and sound. In my opinion, the manuscript deserves publication in Nature Communication due to its pioneering character, scientific quality and in part also as a reference for ubiquitous single-emitter studies in hBN which might be compromised by organic fluorophores (see also <https://arxiv.org/abs/2306.12197>).

Comments:

1) The authors point out the perspective of both-sided encapsulation of molecules by hBN. I also thought this could be a very instructive experiment to carry out. How much of an extra effort would this work add?

Response:

We believe this to be a major effort. One of the problems with stacking layers on top of each other is that contamination and terrylene can cluster into large 'bubbles' or pockets between the two layers, as is well known for heterostructures (S. J. Haigh et al, Nature Materials 11, 2012). We have seen this in our first attempts at stacking hBN layers by polymer-assisted pickup and even exfoliation of hBN onto the silica substrate leads to formation of these bubbles (see for example a bubble in the left corner of Figure S2.2 in Supplementary). Probably, more advanced fabrication techniques, such as polymer-free exfoliation in UHV could help to avoid the formation of these bubbles. However, we are

inexperienced and unequipped for these techniques.

We added a sentence in the caption of Fig. S2.2 to explain how such bubbles may arise and how they make it difficult to encapsulate guest molecules on both sides.

2) Related work on single luminescent nanotubes on hBN (Noe et al, Nano Lett. 18, 4136, 2018) found increased spectral wandering in the proximity of hBN terrace steps, just as they appear in Fig. 1a due to molecule agglomeration. Is there any difference in the spectral characteristics of molecules near the terrace steps and further away? And could the spectral wandering (induced by fluctuating proximal two-level systems) be related to high density of dangling bonds of the hBN terrace edges as opposed to rather low density of vacancy defects on extended terraces?

Response:

We thank the reviewer for drawing our attention to this interesting article, which clearly shows that hBN is a much more stable environment than silica. For the molecules on hBN, we did not observe a clear difference in spectral diffusion near edges or terraces. Also, molecules are not visible on AFM scans, hence we cannot pinpoint with enough precision how far the molecule is removed from an edge. We think this high precision would be required to answer the reviewer's question. Because we observe that the amount and nature of spectral diffusion of different molecules varies dramatically inside the same focal area, we believe that the molecule's coupling to dynamics is very local.

We added a sentence in the introduction comparing hBN to a silica surface, with reference to Noe et al.

3) The blinking behavior as in Fig. 5b should yield strong bunching in auto-correlation measurements as in Fig. 1a and S6.1 on time scales of the characteristic switching time between the "on" and "off" states (as e.g. in Nutz et al, Nano Lett. 19, 7078, 2019). Can the authors expand their time window in the auto-correlation data? This on the one hand might yield a different normalization constant for $g^2(\tau)$, and on the other hand might help to clarify the role of triplet states vs. alternative sources of blinking and thus instructively complement the data in Fig. 5e.

Response:

We have recorded both an antibunching histogram, extended up to 17 μ s, and a fluorescence time trace of 60 seconds with a 10 μ s integration time. We have combined these two data sets to have an extended time window for the autocorrelation function and a non-unity normalization constant in the antibunching histogram. Interestingly, the autocorrelation function reveals a timescale for the other

source of blinking, which was not possible to deduce from the histogram for dark times in Figure 5e.

The new autocorrelation data are now presented in Figure S7.5b, with a reference sentence in the main text.

Reviewer #2

Remarks to the author:

The paper from Smit et al. submitted to Nature Communications details experiments carried out to investigate the optical properties of terrylene molecules deposited on hexagonal boron nitride (hBN). The authors find that molecules can have narrow linewidth transitions even though they are not included inside a protective crystalline matrix. This is significant and novel as molecules on surfaces usually have very broad emission. They find that the molecule emission is more stable when the hBN substrate has been annealed at high temperature. This could be due to the removal of organic contaminants, or some kind of surface passivation. The authors explore the emission spectra of molecules, as well as anti-bunching measurements and laser fluorescence excitation spectra. They compare the molecule emission at various temperatures and observe and analyze broadening effects. Whilst I find the work very interesting, I am unsure whether it is of broad interest nor is it clear whether there are specific applications the authors have in mind which this combination of materials would be useful for. The authors mention “for investigation of physical phenomena on surfaces” which I think they could expand on significantly. They also mention for “nanophotonic devices” but it is not clear how one would combine this hBN with molecules and a waveguide or cavity. The authors also point out that encapsulation in hBN would also be a route to achieving narrower lines, but it is not clear whether this would be inherently better than other more well explored crystalline matrices used to host organic molecules.

Comments:

1. The authors mention that waveguide coupling of molecules was first seen in Shkarin et al (Ref. 15) but previous demonstrations from the same group were made by Türschmann et al., Nano Lett. (2017) and later with ring resonators by Rattenbacher et al., New J. Phys. (2019), as well as demonstrations of coupling to a gap between waveguides by Boissier et al., Nat. Comms. (2021).

Response:

Thank you for pointing us out these works that precede the work of Shkarin et al.

We have now corrected our text, and added the reference to Türschmann.

2. UV and NIR are not defined.

Response:

We agree.

We changed these terms to their unabbreviated form.

3. Was the data in Fig. 2b with annealed or not annealed hBN?

Response:

This figure concerns annealed hBN. The sample was annealed at 750 °C for a period of 12 hours.

We added this information to the caption of Figure 2b.

4. DBT in anthracene temperature broadening has been analyzed in other papers which give activation energies similar to those considered in the text. See Grandi et al., PRA (2016) and Clear et al. PRL (2020).

Response:

We thank the reviewer for pointing out these works.

We have added these references to the text.

5. It would be good to make the red points in Fig. 3 slightly transparent so the distribution of the blue points behind can be seen.

Response:

Indeed.

We made the blue points more visible by turning them into circles of smaller size. We also corrected a mistake in the former figure caption. The amount of statistics mentioned concerns Figure 3a, while the statistics in Figure 3b and 3c are different. We have changed the figure caption to address this.

6. How fast were the scans taken in Fig. 5a?

Response:

The scan rate was 5 seconds per row, taken by 500 data points per line at an integration time of 10 ms per point.

We added this information in the caption of Figure 5a.

7. Figure 5a would benefit from having a plot that sums all the scans and plots than sum vs detuning to elucidate the asymmetry mentioned in the text.

Response:

We agree.

We have added the sum curve beneath Figure 5a to show the averaged line shape of the molecule.

8. Why does Fig. 5d stop before the curve saturates? It would be better to at least plot this further to show the top of the S-curve even if no more data was taken.

Response:

Because of background and power-broadening issues, we could not record reliable data at higher intensities.

We have plotted the rest of the fitted saturation curve to show the S-shape.

Reviewer #3

Remarks to the author:

The paper by Smit et al. reports about the first observation of narrow zero-phonon-lines (ZPL) of single dye molecules adsorbed on the surface of hexagonal boron-nitride (hBN). To conduct the experiments, the authors use the aromatic hydrocarbon terrylene, which over more than two decades has allowed a number of “firsts” in single molecule spectroscopy. The detection of single molecules on a surface under ambient conditions - given sufficient photostability – is relatively straightforward and recently has been shown for the same system, i.e., terrylene on hBN. While in this latter study the very surprising observation has been the unusually high photostability of single terrylene molecules on a surface in the presence of oxygen, in the present study the novel and intriguing result is the occurrence of narrow single molecule ZPLs on a surface. So far, recording of low temperature ZPLs of surface-adsorbed single molecules has remained elusive, mainly because of additional broadening processes and/or increased spectral diffusion. It appears that the peculiarities

of the hBN substrate has given rise to a number of unexpected results, truly remarkable. Smit et al. characterize the single molecules by well-accepted procedures as fluorescence spectra and triplet lifetimes which indicate the presence of terrylene. Single molecule behavior is verified by photon-antibunching as seen in the fluorescence correlation function which also yields the fluorescence lifetime expected for terrylene. Given the numerous emissions reported from hBN, these are important points. The ZPL linewidth – extracted from excitation spectra - is still appreciably broadened with respect to the lifetime limit which seems to be one characteristic of surface-adsorbed molecules. Overall, the authors present novel and interesting findings which may have a strong impact on further single molecule research and for quantum photonic applications. It is to be expected that also other dye molecules will exhibit single molecule ZPLs on hBN, maybe with even more favorable properties. Moreover, such studies may give an answer to the pending question: What are the intrinsic properties of hBN and/or its surface, respectively, which allow the observation of narrow ZPLs?

The paper is well-written, and all relevant information and procedures are given. I suggest publication in Nature Communications after the authors have considered the following questions and comments.

Comments:

In the introduction and at several other instances the authors mention that their results would be the first in which 0-0-ZPLs have been observed for fluorescent molecules on a surface. This statement is certainly correct for the important case of single molecules, but not in general. Indeed, 0-0-ZPLs have been observed by several groups by low temperature spectral hole-burning of surface-adsorbed dye molecules. To give an example (Chem. Phys. Lett. 118 (1991) 179), spectral holes with a width of about 1 GHz have been reported which translates into a ZPL width of 0.5 GHz. While the surface which has been studied was quite different from that of hBN, the ZPL width in this bulk experiment was even narrower than the median given in the present paper. Since (efficient) spectral hole-burning is a process which typically will prevent the detection of stable single molecule ZPLs, the materials used in former experiments will not work for ZPL single molecule detection. Nevertheless, the referee believes that the paper would profit considerably by a critical discussion of the former results, in particular with respect to the general question of linewidth broadening processes on surfaces.

Response:

We thank the reviewer for pointing out this interesting paper, which indeed investigates molecule-surface interactions in a different regime. Indeed, our work concerns molecules adsorbed physically to the hBN surface, whereas the work mentioned by the reviewer concerns a molecule (quinizarin) chemisorbed on a γ -alumina surface. Even though spectral hole burning is not a study on single

molecules, but rather ensembles, we agree with the reviewer that it is an interesting and complementary observation to ours in the context of molecule-surface interactions.

We have added the suggested reference in the introduction, with a sentence of explanation.

The authors have measured the temperature dependence of the 0-0-ZPL width of a single molecule as obtained from emission spectra. As mentioned by the authors, these suffer from experimental resolution limits at low temperature. Have the authors also tried to measure the temperature dependent linewidth as obtained from excitation spectra which would not suffer from resolution limits? Probably, spectral instabilities make this a difficult endeavor, but for some molecules (Figure 5 (d)) even a saturation curve could be taken. Please comment. By the way, I think the saturation curve has not been discussed in the main text.

Response:

For most molecules, the spectral jumps make it difficult to record the linewidth by an excitation spectrum for a range of temperatures, due to spectral jumps outside of the scannable range. For the molecule in Figure 5 it would be possible to record excitation spectra, but we were limited by the laser's scanning range of at most 20 GHz.

We have added a sentence to the main text to discuss the saturation curve.

The authors mention that after annealing considerably less molecules have been found and present several tentative explanations. To the referee's opinion, they also should take into account that the heat treatment (500 K to 900 K) of their samples can easily lead to reactive sites like the well-known dangling bonds (unsaturated valencies). At these reactive sites, terylene can be readily decomposed. Such an explanation would also be in line with the observation of more molecules again after treatment with hexadecane which will saturate the reactive sites.

Response:

We believe the reviewer refers to the ~2 eV defect emission, which they assign to dangling bonds. This emission is typically activated by annealing [doi: 10.1103/PhysRevLett.123.127401]. We think this assignment is still highly debated. Alternatively, a recent study even suggests that this defect could be of organic origin [doi: 10.1021/acsnano.3c02348]. Admittedly, we cannot completely rule out that some reactive sites with dangling bonds are formed or activated during annealing. However, as mentioned in the text, we do speculate that hBN defects might be responsible for the observation of

red-shifted molecules, which also have a slightly different spectrum (Fig. 2). We therefore added a short comment on this point to the main text.

Sentence added to mention the possibility of reactive site at dangling bonds.

In the introduction the authors use micro-eV and MHz for linewidth information. I think it would be better for the reader not to change the units in the introduction.

Response:

The eV units are often used in the solid-state community, e.g. for colored defects in hBN, instead of the MHz units more familiar to spectroscopists. We agree that the paper should be readable by both communities and changed the second sentence in the abstract to include both notations. In the remainder of the work we will use the common frequency notation used in molecular spectroscopy.

We now express linewidth in both units.

REVIEWERS' COMMENTS

Reviewer #1 (Remarks to the Author):

The authors have addressed all my questions and revised the manuscript accordingly. The only concern that came up with the new data in Fig. S7.5 is the normalization of the autocorrelation data: the authors now use $G^{(2)}(\tau)$ and $g^{(2)}(\tau)$ in the figures of the main text and the supplementary information without defining the functions or their relation explicitly. Once this caveat is resolved, the manuscript can be accepted for publication.

Reviewer #2 (Remarks to the Author):

The authors have addressed my comments suitably. I have no further comments.

Reviewer #3 (Remarks to the Author):

The authors have satisfactorily addressed all questions and comments. Accordingly, I recommend publication of the manuscript in Nature Communications.

Hereby we provide a point-by-point response to the referee's comments.

Actions undertaken in response to the reviewer comments are indicated in red.

Reviewer #1

Remarks to the author:

The authors have addressed all my questions and revised the manuscript accordingly. The only concern that came up with the new data in Fig. S7.5 is the normalization of the autocorrelation data: the authors now use $G^{(2)}(\tau)$ and $g^{(2)}(\tau)$ in the figures of the main text and the supplementary information without defining the functions or their relation explicitly. Once this caveat is resolved, the manuscript can be accepted for publication.

Response:

We agree with the referee that the text requires an explicit relation of the antibunching histograms with the autocorrelation data of Figure S7.5b.

We have added the explicit functions for the data (and fit in panel b) in both panels in the figure caption. In addition, we changed the label of the normalized antibunching histogram to $g_{ab}^{(2)}(\tau)$ to distinguish it explicitly from the complete autocorrelation function $g^{(2)}(\tau)$, that includes photon bunching. In all remaining figures in the supplementary that display antibunching histograms, and also Figure 1b, we changed the label to $g_{ab}^{(2)}(\tau)$ as well.

Reviewer #2

Remarks to the author:

The authors have addressed my comments suitably. I have no further comments.

Response:

Thank you for reviewing our article.

Reviewer #3

Remarks to the author:

The authors have satisfactorily addressed all questions and comments. Accordingly, I recommend publication of the manuscript in Nature Communications.

Response:

Thank you for reviewing our article and your recommendation for publication.